# A Mechanism for the Adsorption of 2-(Hexadecanoylamino)Acetic Acid by Smithsonite: Surface Spectroscopy and Microflotation Experiments

**Bin Luo** , **Junbo Liu \*, Quanjun Liu \*, Chao Song, Li Yu, Shimei Li and Hao Lai**

State Key Laboratory of Complex Nonferrous Metal Resources Clean Utilization, Faculty of Land Resources Engineering, Kunming University of Science and Technology, Kunming 650093, China; luobin894@163.com (B.L.); song_4214_14_sc@163.com (C.S.); 317905202@163.com (L.Y.); luobin806727801@126.com (S.L.); kjlaihao@163.com (H.L.)

**\*** Correspondence: nwenlin@kmust.edu.cn (J.L.); luobin894@kmust.edu.cn (Q.L.)

**Abstract:** Zinc is mostly extracted from oxidized zinc and zinc sulfide minerals, and this process involves flotation as a key step. While it is easier to float the sulfide mineral, its consumption and depletion has led to an increased reliance on oxidized zinc minerals, including smithsonite. Hence, the development of efficient ways of collecting smithsonite by flotation is an important objective. Herein, we describe the use of 2-(hexadecanoylamino)acetic acid (HAA), a novel surfactant, as a collector during smithsonite flotation. The mechanism and flotation performance of HAA during smithsonite flotation was investigated by total organic carbon (TOC) content studies, zeta potential measurements, Fourier-transform infrared (FTIR) spectroscopy, and X-ray photoelectron spectroscopy (XPS) analyses, combined with microflotation experiments. The flotation results revealed that HAA was an excellent collector in pulp over a wide pH range (9–12) and at a relatively low concentration ($2 \times 10^{-4}$ mol/L), at which a recovery of close to 90% of the smithsonite mineral was obtained. TOC content studies revealed that the good flotation recovery was ascribable to large amounts of collector molecule adsorbed on the smithsonite surface, while zeta potential measurements showed that the HAA was chemically adsorbed onto the smithsonite. FTIR and XPS analyses revealed that the HAA collector molecules adsorbed onto the smithsonite surface as zinc–HAA complexes involving carboxylate moieties and Zn sites on the smithsonite surface in alkaline solution.

**Keywords:** smithsonite; flotation; 2-(hexadecanoylamino)acetic acid; collector; adsorption

## 1. Introduction

Oxidized zinc and zinc sulfide ores are the primary naturally occurring zinc ores, and these ores are mainly processed by flotation [1,2]. However, due to the consumption and depletion of zinc sulfide ores, oxidized zinc ores, which are more difficult to float, are being increasingly relied upon. Consequently, the development of methods for the efficient processing of zinc oxide ores has become a hot research topic in recent years [3–5].

As mentioned above, the oxidized mineral is more difficult to float than the sulfide mineral, due to the greater solubility of the surface of the oxidized mineral than that of the sulfide mineral in solution [6,7]. Consequently, collector molecules are unable to adsorb onto the surface of the oxidized mineral in a stable manner [8]. According to the literature, direct flotation and sulfurized flotation are the main methods used to process oxidized zinc minerals. During sulfurized flotation, the oxidized zinc mineral is pre-treated with sodium sulfide, an activator, prior to flotation, which transforms the surface properties of the oxidized mineral into those of the sulfide mineral [3,9,10]. However, there are

some problems associated with the sulfurized flotation process, such as the lack of activity on the surface of oxidized zinc mineral in the absence of $Cu^{2+}$, the low recovery of zinc metal obtained when xanthates are used as collectors, and the flotation foam, which is difficult to eliminate when amines are used as collectors [5,11].

In addition, fatty acids have been used as collectors for the direct flotation of oxidized zinc minerals. Fatty acids are unsatisfactory collectors for industrial use due to their low selectivity, low solubility, and sensitivity to calcium and magnesium ions in the pulp [12]. Hence, the discovery and development of a novel and efficient collector for oxidized zinc minerals flotation is urgently required.

Smithsonite ($ZnCO_3$), a typical oxidized zinc mineral, was used in this work. Smithsonite is also a semisoluble salt mineral: It has a solubility product constant of $1.46 \times 10^{-10}$ M [7]. The main characteristics of a salt mineral are ionic bonding and moderate water solubility [7,13]. The dissolution of smithsonite in solution is described as follows [8]:

$$ZnCO_3 \Leftrightarrow Zn^{2+} + CO_3^{2-} \tag{1}$$

In a flotation system, water molecules chemically adsorb onto the smithsonite surface in water. Compared to zinc sulfide minerals, the main factor responsible for the low natural floatability of smithsonite is the high activity of water dipoles [6,14]. This phenomenon decreases the probability of effective reagent adsorption onto the smithsonite surface. The chemistry of smithsonite dissolution and hydrolysis reactions involving $Zn^{2+}$, together with equilibrium constants in open solution, are shown in Table 1 [8,15,16].

**Table 1.** Pertinent reactions and thermodynamic data for smithsonite dissolution in open solution.

| Pertinent Reaction and Constant | | Pertinent Reaction and Constant | |
|---|---|---|---|
| (a) $CO_3^{2-} + H^+ \Leftrightarrow HCO_3^-$ | 10.33 | (h) $Zn(CO_3)_{0.4}(OH)_{1.2} \Leftrightarrow Zn^{2+} + 0.4CO_3^{2-} + 1.2OH^-$ | 14.85 |
| (b) $CO_2(g) + OH- \Leftrightarrow HCO_3^-$ | 6.18 | (i) $Zn^{2+} + 4OH^- \Leftrightarrow Zn(OH)_4^{2-}$ | 14.80 |
| (c) $OH^- + H^+ \Leftrightarrow H_2O$ | 14.00 | (j) $Zn^{2+} + HCO_3^- \Leftrightarrow ZnHCO_3^+$ | 2.10 |
| (d) $HCO_3^- + H^+ \Leftrightarrow H_2CO_3$ | 6.35 | (k) $Zn^{2+} + CO_3^{2-} \Leftrightarrow ZnCO_3$ | 5.30 |
| (e) $Zn^{2+} + OH^- \Leftrightarrow ZnOH^+$ | 5.00 | (l) $Zn^{2+} + 2CO_3^{2-} \Leftrightarrow Zn(CO_3)_2^{2-}$ | 9.63 |
| (f) $Zn^{2+} + 2OH^- \Leftrightarrow Zn(OH)_2(aq)$ | 11.10 | (n) $Zn^{2+} + CO_3^{2-} \Leftrightarrow ZnCO_3(s)$ | 10.00 |
| (g) $Zn^{2+} + 3OH^- \Leftrightarrow Zn(OH)_3^-$ | 13.60 | | |

The flotation agent used in this study, 2-(Hexadecanoylamino)acetic acid (HAA), is currently mainly used in the commercial cosmetics and medicine fields. Rimmerman et al. [17] have shown that HAA is a novel endogenous lipid that acts as a modulator of calcium influx and nitric oxide production in sensory neurons, whereas Sabrina et al. [18] have reported that HAA is a histidine-based amino acid that can be used as a gel to increase the skin permeation of metronidazole. HAA is a novel surfactant that may be suitable for the collection of smithsonite. However, HAA has not yet been used to collect smithsonite during flotation: Hence, its flotation behavior and mechanism for adsorption onto the smithsonite surface are unknown. Therefore, it is important to study and discuss the flotation behavior and adsorption mechanism of HAA during smithsonite flotation. The mechanism and flotation performance of HAA adsorbed onto the smithsonite surface was investigated by microflotation experiments, total organic carbon (TOC) content studies, zeta potential measurements, Fourier-transform infrared (FTIR) spectroscopy, and X-ray photoelectron spectroscopy (XPS).

## 2. Experiments

### 2.1. Materials and Reagents

The pure smithsonite samples used in all experiments were obtained from Yunnan Province in China. Subsequently, the samples were crushed and dry-ground using an agate mortar and pestle. For microflotation tests and TOC content studies, the ground product of smithsonite was sieved

to achieve a particle fraction of −74 + 38 μm using a standard screen. The rest of the sample was ground to a particle fraction finer than 5 μm for zeta potential measurements, FTIR, and XPS analyses. The results of X-ray diffraction (XRD) patterns are shown in Figure 1, where it can be seen that only the diffraction peak of smithsonite was detected. It was confirmed that the smithsonite samples used in the experiments were of high purity.

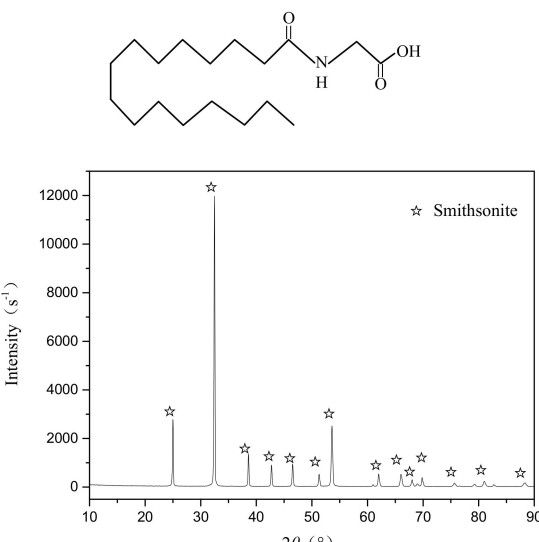

**Figure 1.** X-ray diffraction pattern of the pure smithsonite sample.

AR-grade sodium chloride (NaCl), hydrochloric acid (HCl), and sodium hydroxide (NaOH) were purchased from the Tianjin reagent chemical company of China. Deionized water (DI water) was used in all experiments. The collector HAA was purchased from Shanghai Aladdin Bio-Chem Technology Co., Ltd., Shanghai, China. The structure of HAA is shown as follows.

## 2.2. Microflotation Experiments

Microflotation tests were implemented under mechanical agitation in a 50-mL flotation cell. Four grams of pure smithsonite sample were added to 40 mL of solution in all experiments. HCl and NaOH were used to adjust the pulp pH. The smithsonite samples were conditioned with HAA for 3 min in required concentrations, and terpenic oil was used as frother at a concentration of 0.2 g/L. They floated for 3 min with an aeration rate of 4 L/min. The stirring speed was 1400 r/min during the flotation. After the flotation tests, the tailings and concentrates were weighed separately after filtration and drying. The recovery was calculated based on the solid mass distribution between the float concentrate and tailings. All of the microflotation experiments were completed in triplicate. The final experimental results showed the average values.

## 2.3. TOC Content Study

Four grams of smithsonite powder (−74 + 38 μm) were introduced into 40 mL of collector solution with the desired concentration at pH 9. After stirring for 20 min, the solution was left to sit for 1 h. Subsequently, the suspension was filtered, and the liquor samples were taken for TOC measurements with the equipment HTY-DI1500 (Zhejiang Tailin Bioengineering Co., Ltd., Hangzhou, China). Both of the TOC amounts of the original collector solutions and treatment samples were tested to obtain the TOC reduction contribution to smithsonite adsorption. All of the samples were detected in triplicate, and the results were shown in the form of average values.

## 2.4. Zeta Potential Measurements

Zeta potential was measured in $1 \times 10^{-3}$ mol/L NaCl background electrolyte solution using a zeta potential analyzer (Zetasizer Nano S90, Malvern Instruments Ltd., Malvern, United Kingdom). Here, 0.1 g of $-5$ μm smithsonite was added to 100 mL NaCl background electrolyte solution. After stirring for 3 min at the desired reagent concentration at various pH values, the pulp sat for 5 min. The zeta potential was measured after about 10 mL of the supernatant liquid was transferred into the measurement cell. All of the experiments were conducted at room temperature. Each sample was measured in triplicate during zeta potential measurements, and experimental results showed their average values.

## 2.5. FTIR Spectrum

FTIR was conducted with an Avatar 300 from the Thermo Electron Co., Cridersville, America. The FTIR spectra in the range from 400 to 4000 cm$^{-1}$ were recorded. About 0.2 g of $-5$ μm smithsonite powder was added to 100 mL collector solutions at a concentration of $3 \times 10^{-4}$ mol/L. The pH of the suspension was maintained at 9 through the addition of the dilute NaOH solution during the condition. After stirring for 30 min and then sitting for 2 h, the smithsonite sample was gently washed five times with DI water at pH 9 and air-dried. Fifty milligrams of smithsonite powder were mixed with 100 mg of KBr powder in an agate. The mixture was ground until it was thoroughly mixed. The powdered mixture was then pressed into a thin plate for FTIR analysis.

## 2.6. XPS Analysis

XPS experiments were conducted with a PHI 5000 Versa Probe II (PHI5000, ULVAC-PHI, Tokyo, Japan) equipped with an Al target. Smithsonite samples ($-5$ μm) were treated with collector solution at a concentration of $3 \times 10^{-4}$ mol/L. The pH of the solution was adjusted to 9. Then, there was stirring for 30 min and sitting for 2 h. After the adsorption achieved equilibrium, the suspension was filtered and air-dry-prepared for XPS determination. Subsequently, MultiPak Spectrum software was used to analyze the spectra and calculate surface atomic ratios of the measured samples. All the spectra were adjusted on the basis of the standard C1s binding energy (284.8 eV) and were further fitted by the Gauss–Lorentz method.

## 3. Results and Discussion

### 3.1. Flotation Study

Smithsonite is zinc carbonate ($ZnCO_3$), and carbonate minerals are well known to react readily under acidic conditions to consume large amounts of acid. Therefore, all experiments in this study were conducted only under neutral and alkaline pH conditions. Figure 2 displays the flotation recovery of smithsonite as a function of pulp pH (6–12) when treated with the collector at a concentration of $2 \times 10^{-4}$ mol/L. The results clearly reveal that floatability increased as the pH increased from 6 to 9. No further increases in recovery were observed at higher pH values, at which the recovery remained relatively stable. In other words, the optimal flotation recovery of smithsonite was achieved at pH 9.

To further know the influence of the HAA collector on smithsonite flotation, microflotation work was beneficial in seeing a standard gangue mineral (dolomite) to understand the selectivity of the reagent for smithsonite flotation. Recovery of smithsonite and dolomite as a function of collector concentrations was conducted at pH 9. Figure 3 shows excellent selectivity of HAA for smithsonite flotation in low collector concentration solution. The increasing collector dosage improved the recovery both of smithsonite and dolomite. Smithsonite recovery dramatically increased as the collector concentration was varied from $1.5 \times 10^{-4}$ to $2 \times 10^{-4}$ mol/L. Recovery was observed to increase only slightly at even higher concentrations. In particular, the flotation recovery of smithsonite exceeded 90%, which is a desired indicator, at collector concentrations in excess of $2 \times 10^{-4}$ mol/L. Dolomite showed poor floatability in low HAA concentrations, and the floatability increased with the

increasing collector concentrations. However, the recovery of dolomite did not reach 40% when the collector concentrations increased to $3 \times 10^{-4}$ mol/L.

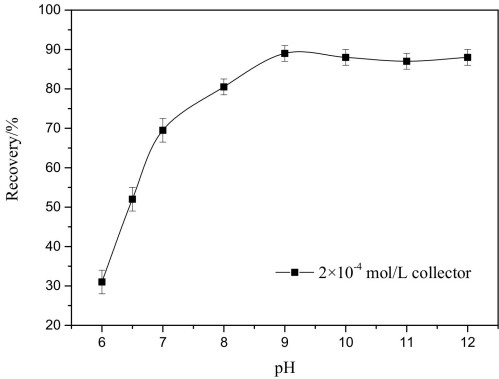

**Figure 2.** Recovery of smithsonite as a function of pH.

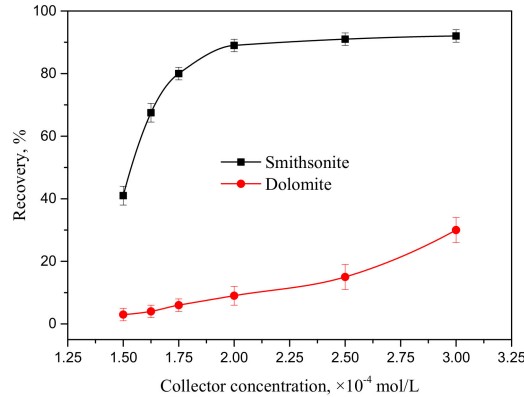

**Figure 3.** Recovery of smithsonite and dolomite as a function of collector concentrations at pH 9.

The above flotation analyses indicate that HAA was an excellent collector for smithsonite flotation. The floatability of smithsonite in the alkaline slurry was better than neutral pH conditions using HAA as the collector. According to the results of the pH flotation study, pH 9 was significant, since excellent flotation and recovery of smithsonite were observed at that value. Hence, all subsequent studies, including TOC content determination, zeta potential measurements, FTIR spectroscopy, and XPS analyses were performed at pH 9.

### 3.2. HAA Adsorbance on the Smithsonite Surface

In order to obtain more information about the adsorption state of the collector on the smithsonite surface, the amount of adsorbed collector was calculated from the difference in the TOC content of the solution before and after treatment of the smithsonite. Figure 4 displays the amount of adsorbed collector as functions of collector concentrations. At a low collector concentration, the amount of adsorbed HAA was relatively low. The adsorption of the collector increased faster as the collector concentration was increased from $1.5 \times 10^{-4}$ to $2 \times 10^{-4}$ mol/L. The change in TOC content was not as significant at concentrations in excess of $2 \times 10^{-4}$ mol/L, and the amount of adsorbed collector gradually stabilized at concentrations greater than $2.5 \times 10^{-4}$ mol/L. This trend was consistent with the results from the flotation experiments. The excellent floatability of smithsonite was attributable to large amounts of collector molecule adsorbed onto the smithsonite surface.

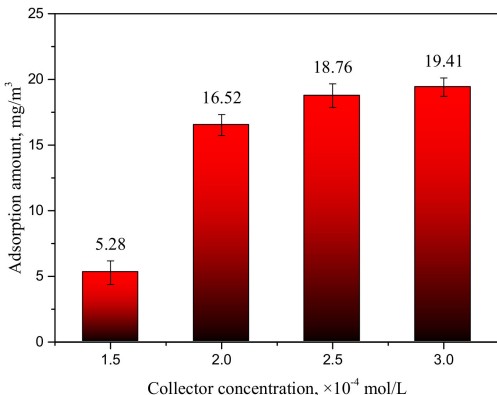

**Figure 4.** The amount of adsorbed collector as a function of collector concentration at pH 9.

### 3.3. Effect of HAA on the Zeta Potential of Smithsonite

Samples of smithsonite powder were placed in a zeta potential cell, either in the absence or presence of the collector. Following instrument calibration, the zeta potential of the smithsonite sample was determined in order to provide information on the collector-ion species in the pulp suspension that was adsorbed on the smithsonite surface.

The dissolution species of smithsonite is complex. Some studies have shown that $\equiv ZnOH^0$ and $\equiv CO_3H^0$ (where "$\equiv$" represents the surface) are the two primary hydration sites that govern surface speciation [15,16]. There is a large amount of experimental evidence indicating that complex reactions occur on the surface of smithsonite. According to the literature, some of the reactions that occur on the surface of smithsonite depend on the solution conditions: These reactions are described as follows [19–21]:

$$\equiv CO_3H^0 \Leftrightarrow \equiv CO_3{}^- + H^+ \tag{2}$$

$$\equiv CO_3H^0 + Zn^{2+} \Leftrightarrow \equiv CO_3Zn^+ + H^+ \tag{3}$$

$$\equiv CO_3H^0 + ZnOH^+ \Leftrightarrow \equiv CO_3ZnOH^0 + H^+ \tag{4}$$

$$\equiv ZnOH^0 + H^+ \Leftrightarrow \equiv ZnOH_2{}^+ \tag{5}$$

$$\equiv ZnOH^0 \Leftrightarrow \equiv ZnO^- + H^+ \tag{6}$$

$$\equiv ZnOH^0 + HCO_3{}^- + H^+ \Leftrightarrow \equiv ZnHCO_3{}^0 + H_2O \tag{7}$$

$$\equiv ZnOH^0 + CO_3{}^{2-} + H^+ \Leftrightarrow \equiv ZnCO_3{}^- + H_2O \tag{8}$$

As is evident from the above, the smithsonite surface is charged through non-stoichiometric dissolution and hydrolysis, with some ions released into the solution. The $Zn^{2+}$ hydrolysis products then readsorb onto the smithsonite surface. The smithsonite surface charge is also determined by interactions between ions in its crystal structure and the hydrolysis products. Some ions ($Zn^{2+}$, $HCO_3{}^-$, $H^+$, $OH^-$, and $CO_3{}^{2-}$) determine the potential between the smithsonite surface and the solution, and deprotonation and protonation reactions determine the charge of the smithsonite surface: These reactions depend on the solution conditions.

Figure 5 displays the zeta potentials of smithsonite in the absence and presence of the collector as functions of pH. The zeta potential of pure smithsonite became more negative with increasing pH, which was consistent with the findings of former investigations [5,22]. The isoelectric point (IEP) of smithsonite has been reported to lie between pH 7 and 8.8 [8,20]. As is evident from Figure 5, the IEP of pure smithsonite in the absence of the collector was approximately pH 7.6, which was in agreement with the results of previous studies. The zeta potential of smithsonite rapidly declined at pH values above 7.6.

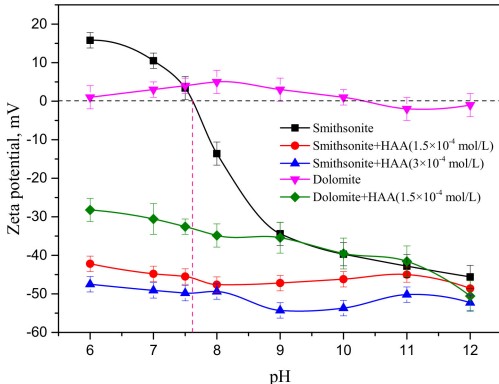

**Figure 5.** Zeta potential of smithsonite as a function of pH after conditioning with deionized (DI) water and the collector.

The change in the potential of the surface of pure smithsonite has previously been described in the following way [3,5,8,19,20]: The increasing zeta potential of pure smithsonite (Figure 5), from 0 mV at pH 7.6 to 15.80 mV at pH 6, is ascribable to the chemical adsorption of protons and $Zn^{2+}$ according to Equations (2) and (3), which results in increasing concentrations of $Zn^{2+}$ and hydrogen ions in solution with decreasing pH, and the smithsonite dissolves as a consequence. According to the distributions of species in previous studies, the concentration of $Zn^{2+}$ increases faster than that of hydrogen ions. At the same time, the $\equiv ZnOH_2^+$ site on the mineral surface (Equation (5)) is formed by the protonation of $\equiv ZnOH^0$ [23,24]. As the solution pH is increased from 7.6 to 9, the zeta potential reveals that the smithsonite becomes negatively charged due to the deprotonation of $\equiv ZnOH_2^+$, and two sites ($\equiv ZnHCO_3^0$ and $\equiv ZnCO_3^-$) are formed through the chemical adsorption of $HCO_3^-$ and $CO_3^{2-}$ (Equations (7) and (8)) [8]. Meanwhile, the chemical adsorption of $ZnOH^+$ is possible at the $\equiv CO_3H^0$ site (Equation (4)) [25,26]. At pH > 9, the slope of the zeta potential curve for pure smithsonite gradually flattens, as opposed to the curve at 7.6 < pH < 9, which indicates that the hydrolysis products may have readsorbed onto the mineral surface, resulting in a more negative zeta potential at a pH greater than 9. According to previous investigations, the $\equiv ZnO^-$ species are dominant in solutions at pH > 12, which confirms that Equation (6) rarely takes place under these solution conditions.

Figure 5 reveals that the zeta potential of smithsonite and dolomite became more negative in the collector solution compared to that of untreated samples. Furthermore, higher collector concentrations led to more negative charge on the mineral surface. The HAA collector may have ionized to form an anionic collector in solution due to the presence of the carboxylic acid group in the HAA molecule: The increase in the degree of ionization would be expected to follow the rise in solution alkalinity. Therefore, the anionic collector molecules were absorbed onto the mineral surface, resulting in a decrease in the zeta potential of smithsonite and dolomite in the 6–12 pH range. All of these experiments showed that the zeta potentials were more negative than those obtained in DI water, in which collector ions were unable to physically adsorb onto the smithsonite surface due to electrostatic repulsion. Therefore, the collector became chemisorbed onto the smithsonite surface.

### 3.4. FTIR Spectroscopy

In order to gain further insight into the adsorption state of the collector on the mineral surface, we subjected the HAA collector, before and after absorption by smithsonite, to FTIR spectroscopy, the results of which are shown in Figure 6.

Spectrum (a) is that of HAA in the 500–4000 $cm^{-1}$ range: This spectrum exhibited peaks at 3327 $cm^{-1}$ due to the stretching vibrations of the N–H bond, at 2918 and 2849 $cm^{-1}$ that corresponded to the C–H stretches of $-CH_3$ and $-CH_2$ units, and at 1703 $cm^{-1}$, which was assigned to the C=O bond of the carboxylic acid moiety in the collector molecule [27,28]. The peaks at 1645 $cm^{-1}$ and 1559 $cm^{-1}$ were assigned to amide I and II vibrations: Amide I arose from the C=O stretch, while the amide II

band was mostly due to N–H and C–H vibrations [29–32]; peaks that appeared at 1472 and 719 cm$^{-1}$ were representative of C–H bending; and C–N and C–O stretching bands were observed at 1272 and 1037 cm$^{-1}$, respectively [33–35]. In addition, spectra (b) and (c) belong to smithsonite samples treated with DI water and the HAA collector, respectively. Spectrum (b) exhibits three strong peaks at 1428, 870, and 743 cm$^{-1}$, which were characteristic smithsonite bands that corresponded to the asymmetric stretching (C–O) and bending (C–O–C) vibrations of CO$_3{}^{2-}$ [36,37]. Meanwhile, the obvious peak at 3448 cm$^{-1}$ was attributed to the O–H stretching vibrations of the water molecule present on the mineral surface as a consequence of exposure to DI water [34,38]. Compared to spectrum (b), some new peaks were observed in spectrum (c) of the smithsonite sample following collector treatment, namely new peaks at 2918 and 2849 cm$^{-1}$ that corresponded to hydrocarbon chains (–CH$_3$ and –CH$_2$ bond), 1617 cm$^{-1}$ (C=O bond in amide), and 1558 cm$^{-1}$ (N–H bond in amide). The broad peak observed at 3435 cm$^{-1}$ in spectrum (c) was due to overlapping N–H and O–H stretching peaks following collector treatment. Meanwhile, the peak at 1703 cm$^{-1}$ disappeared upon collector treatment. Its disappearance, and the rise of peaks at 3435, 1617, and 1558 cm$^{-1}$, were further evidence that chemical adsorption on the smithsonite surface occurred through Zn sites that reacted with carboxyl groups of HAA molecules. The characteristic peaks for the C–O–C and C–O in CO$_3{}^{2-}$ units of smithsonite following absorption of the HAA collector did not significantly shift from their positions prior to treatment, which confirmed that the CO$_3{}^{2-}$ groups did not participate in the chemical reaction.

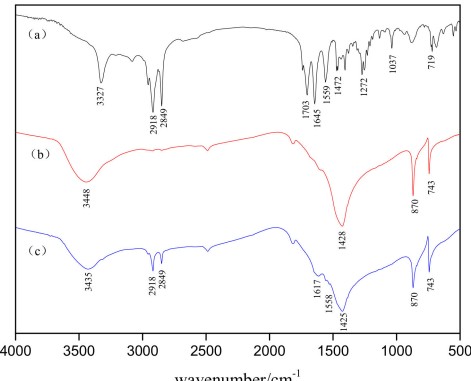

**Figure 6.** Fourier-transform infrared (FTIR) spectra of (**a**) the 2-(hexadecanoylamino)acetic acid (HAA) collector, and (**b**,**c**) smithsonite before and after treatment with HAA (concentration: $5 \times 10^{-4}$ mol/L) at pH 9, respectively.

These FTIR analyses indicate that the HAA collector was successfully adsorbed by the smithsonite, most likely through interactions of its carboxyl group with Zn sites on the smithsonite surface in weakly alkaline solutions, possibly through the formation of covalent bonds.

*3.5. XPS Analyses*

In order to confirm the interpretation of the FTIR spectra and to further understand the adsorption state of the smithsonite surface during flotation, smithsonite samples were analyzed before and after treatment with HAA by XPS. The C1s, O1s, Zn2p, and N1s binding energies were then determined by spectral peak fitting, and MultiPak Spectrum software was used to calculate the relative concentrations of the various elements on the smithsonite surface in the absence and presence of the collector. Figure 7 shows full XPS spectra of smithsonite with and without collector treatment. Figure 8 displays the XPS spectra processed by the MultiPak Spectrum software. Panels A-1, B-1, C-1, and D-1 show the XPS spectra of a pure smithsonite sample treated with DI water. Panel A-1 shows the full spectrum, while the C1s, O1s, and Zn2p spectra are displayed in panels B-1, C-1, and D-1, respectively. Panels A-2, B-2, C-2, and D-2 are the corresponding spectra of the smithsonite samples treated with the HAA collector, while panel E-2 shows the N1s spectrum of smithsonite following collector adsorption.

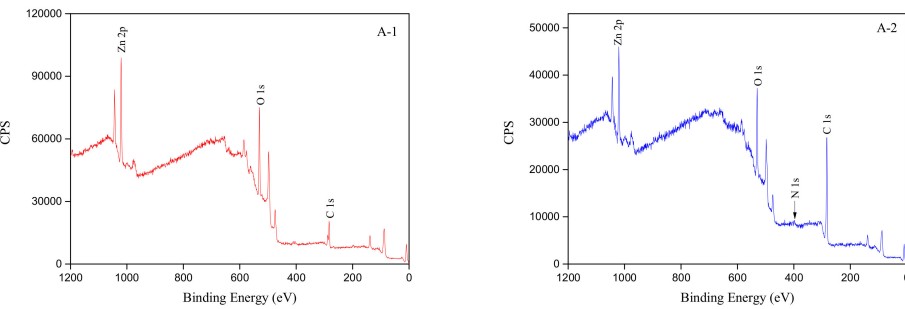

**Figure 7.** Full X-ray photoelectron spectroscopy (XPS) spectra of smithsonite treated with DI water (left) and collector (right, concentration of $5 \times 10^{-4}$ mol/L) at pH 9.

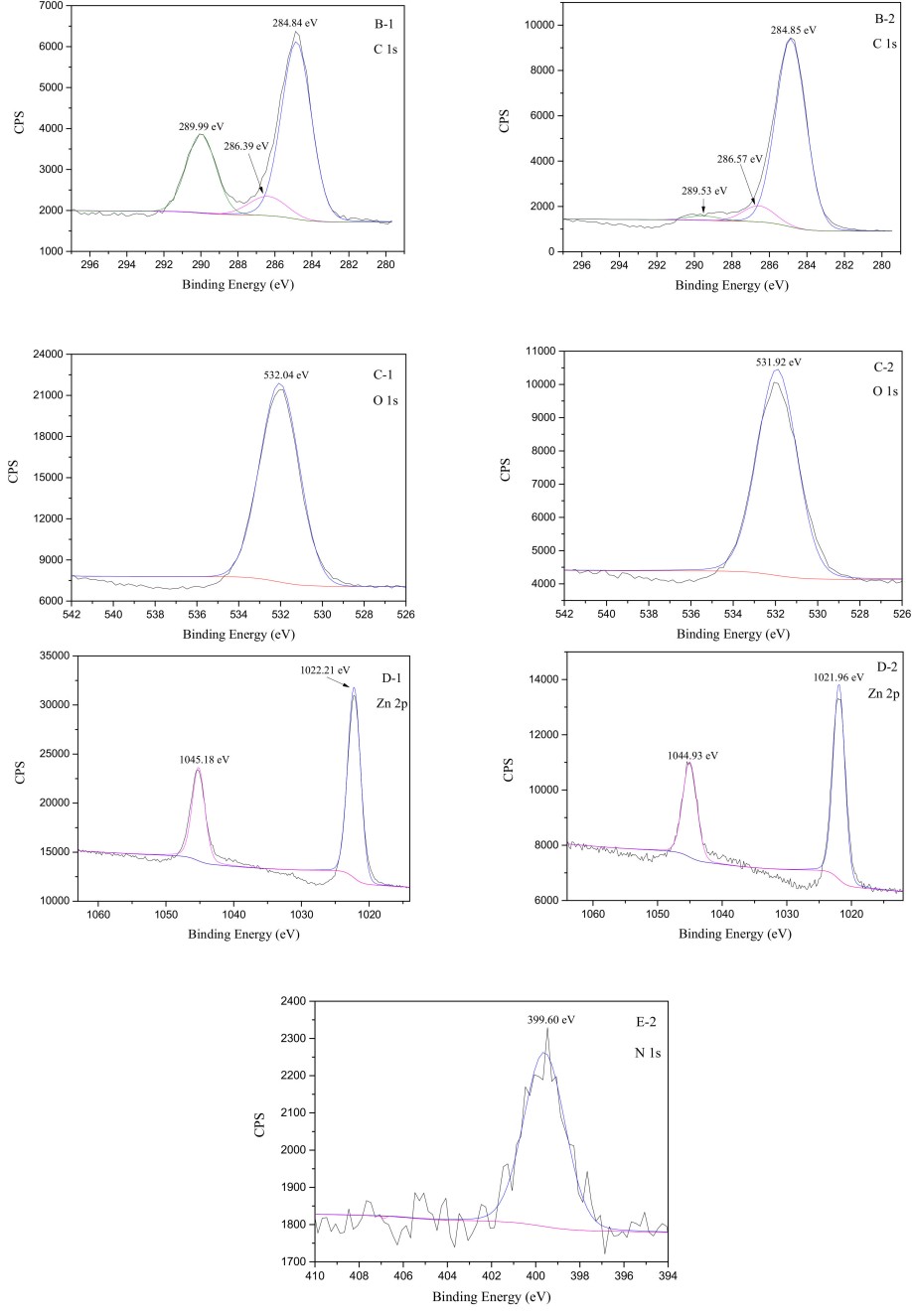

**Figure 8.** XPS spectra of smithsonite treated with DI water (left) and collector (right, concentration of $5 \times 10^{-4}$ mol/L) at pH 9.

Panel A-1 in Figure 7 reveals that only elemental C, O, and Zn were detected in the full XPS spectrum of pure smithsonite treated with DI water, which shows that the smithsonite samples used in these experiments were of high purity. Compared to panel A-1, an N signal was observed in the XPS spectrum (panel A-2) after the smithsonite sample was treated with the collector, which was consistent with adsorption of the collector onto the smithsonite surface. Table 2 provides a semiquantitative summary of the atomic concentrations of the smithsonite surface before and after treatment with the collector. The carbon concentration was observed to increase from 42.38% to 69.59%, while the oxygen concentration decreased from 39.08% to 24.40%, and the zinc concentration decreased from 15.53% to 7.38%. At the same time, nitrogen, at a level of 2.97%, emerged on the mineral surface following treatment with the collector. These changes in atomic concentrations are further evidence that the collector was adsorbed onto the mineral surface.

**Table 2.** The results of atomic concentration on the smithsonite surface by semiquantitative analysis.

| Samples | Atomic Concentration, % | | | |
|---|---|---|---|---|
| | C1s | O1s | Zn2p | N1s |
| Raw material | 42.38 | 39.08 | 15.53 | - |
| Collector adsorption | 69.59 | 24.40 | 7.38 | 2.97 |

In panel B-1, the two C1s peaks, at 286.39 and 289.99 eV, corresponded to C–O and C=O bonds assigned to the carbon in the carbonate group of smithsonite samples [37,39], while the other peak at 284.84 eV was attributed to C–C and C–H contaminants [3]. A 0.18 eV shift in the C–O-bond peak and a 0.46 eV change in the energy of the C=O bonds in the carbonate groups shown in panel B-2 were attributed to collector adsorption. At the same time, the binding energy at 289.53 eV shown in panel B-2 was attributed to the C=O bond in the carboxylic group of the collector and carbonate groups [37]. In other words, only the C1s peaks at 289.99, 286.39, 289.53, and 286.57 eV were derived from the smithsonite sample: The remaining C1s binding energies were not assigned to the surface composition of pure smithsonite. In panel C-1, the O1s binding energy at 532.04 eV corresponded to the O in the C–O and C=O bonds of the carbonate group [37,40]. The O1s binding energy shifted from 532.04 to 531.92 eV following collector adsorption onto the smithsonite surface. In panel D-1, the two peaks at 1045.18 and 1022.21 eV in the spectrum of the untreated smithsonite sample corresponded to $Zn2p_{1/2}$ and $Zn2p_{2/3}$ binding energies and were consistent with previous reports [3,16,37,40,41]. Following collector adsorption onto the smithsonite, the $Zn2p_{1/2}$ binding energy was observed to shift to 1044.93 eV, while the $Zn2p_{2/3}$ energy moved to 1021.96 eV: These 0.25 eV differences were attributable to Zn–O bond formation according to previous studies [37], which was consistent with chemical adsorption of the collector at the Zn sites on the smithsonite surface. The carboxylic acid was deprotonated in alkaline solution. The formation of Zn–O bonds and the structure of the collector suggest that Zn–O bonds were formed through reactions of the Zn sites on the smithsonite surface and the carboxylates of deprotonated collector molecules. According to former studies, the first N1s peak at 399.60 eV (panel E-2) is attributed to the amino group of the HAA collector [42,43].

In a word, changes in the O1s, Zn2p, and N1s binding energies indicate that the HAA collector was chemically adsorbed onto the smithsonite surface. According to the XPS results, the HAA collector was adsorbed onto the smithsonite surface through Zn sites that reacted with the carboxylate of the deprotonated HAA to form covalent bonds, which was consistent with the FTIR observations. In other words, the collector molecules were adsorbed on the smithsonite surface as zinc–HAA complexes. The proposed adsorption of the HAA molecule onto the smithsonite surface is shown Figure 9.

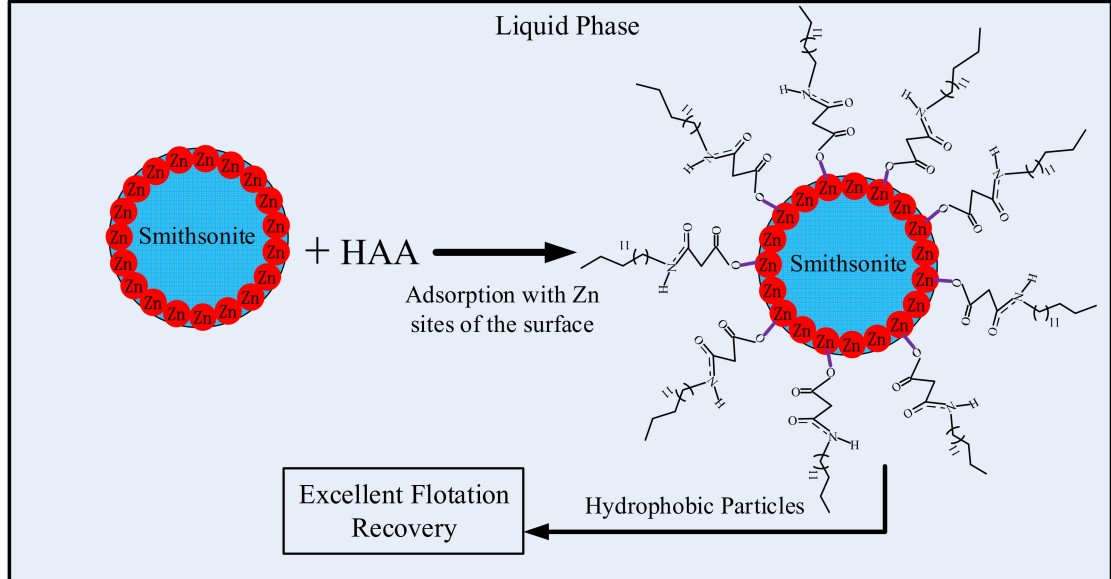

**Figure 9.** Depicting the proposed process for the adsorption of HAA molecules on the smithsonite surface.

## 4. Conclusions

HAA surfactant was used for the first time as a collector for smithsonite flotation. HAA exhibited excellent collecting performance, as evidenced by microflotation experiments and TOC content studies. The flotation recovery of pure smithsonite was 90% at a collector concentration of $2 \times 10^{-4}$ mol/L over a wide range of pulp pH values (9–12). According to the TOC content studies, the good flotation recovery was ascribable to large amounts of collector molecule adsorbed on the smithsonite surface.

Zeta potential measurements revealed that the HAA was chemically adsorbed onto the smithsonite. The disappearance of the carboxyl peak in the FTIR spectrum and the observed shift in the position of the C=O peak of the amide group indicate that HAA was successfully adsorbed onto the smithsonite surface, most likely through reactions of carboxylate moieties with Zn sites on the smithsonite surface. XPS further revealed that the HAA collector was adsorbed on the smithsonite surface through the formation of covalently bound zinc–HAA complexes, in which HAA molecules interacted with Zn sites on the smithsonite surface.

**Author Contributions:** B.L. and Q.L. conceived and designed the experiments; B.L. and J.L. performed the experiments; C.S., L.Y., S.L. and H.L. analyzed the data; B.L. wrote this paper and J.L. corrected it.

**Funding:** The research project was financially supported by the National Natural Science Foundation of China (No.51764021) and the Analysis and Testing Foundation of Kunming University of Science and Technology (No.2017M20152201097).

**Conflicts of Interest:** The authors declare no conflict of interest.

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
