# Peer review of "A Mechanism for the Adsorption of 2-(Hexadecanoylamino)Acetic Acid by Smithsonite: Surface Spectroscopy and Microflotation Experiments"

_minerals, doi:10.3390/min9010015_

Round 1
Reviewer 1 Report
The authors have presented their findings describing a new collector for smithsonite flotation. The manuscript should be accepted for publication pending the following required revisions:
-The authors should elaborate on the claim in the second paragraph of the introduction that oxide mineral surfaces are inherently more reactive than sulfides. Supporting references should be provided
-The authors should not use vulcanizing here as I believe the correct term in terms of flotation is sulfidizing
-The formula for smithsonite should be provided in the introduction rather than later in the manuscript
-"Agate torsion mortar" should be "agate mortar and pestle"
-The authors indicate that HAA was purchased from a biochem technology company. Please indicate in the manuscript what the current commercial uses of this reagent are
-The authors mention that flotation, TOC measurements and zeta potentials were all conducted in triplicate. Standard deviations must be included to provide an indication of the repeatability of these measurements
-For the TOC study the work would benefit from a baseline measurement of smithsonite absent any collector. It would seem like the semi-soluble carbonate nature of this mineral might lead to a baseline reading separate from the collector carbon
-For the flotation results as a function of collector concentration it would be beneficial to discuss the collector addition in terms of moles of reagent per surface area of mineral (measured by BET or similar technique)
-For the TOC results in Figure 4 did the authors consider preparing an adsorption isotherm to provide further indications into the type and density of collector adsorption onto the mineral surface?
-The microflotation work would be beneficial to see vs a standard gangue mineral to understand the selectivity of the reagent for smithsonite. The same comment follows for zeta potential measurements
-The manuscript would be strengthened by the inclusion of a speciation diagram in the discussion of the complex reactions occurring at the mineral surface as a function of pH
-In the zeta potential sample preparation the authors employed NaCl as the background electrolyte rather than KCl. Does this have any impact on mineral solubility or interactions with carbonate ions in solution?
-The authors should mention that zeta potential results may be very different at higher solids concentrations present in industrial flotation cells as the concentrations of dissolved Zn, carbonate, and other potential determining ions will be higher
-The authors need to include the structure of HAA prior to the FTIR and XPS discussions
-The manuscript would be further strengthened by bench flotation of smithsonite from a real ore system with typical gangue minerals present
Author Response
Reply to the Review Report is shown at PDF file

Reviewer 2 Report
In this paper, the authors present a novel reagent for the flotation recovery of Smithsonite. This is of interest as easy to process sulphide ores are being depleted at a rapid pace.
The experimental design is adequate for answering the research questions and below are suggestions for further improving the quality of the paper.
1.0 Introduction
The authors provide an adequate explanation of the importance of the recovery of Smithsonite to justify the research. There is, however, some confusing statements as it appears that the authors use zinc oxide and oxidised zinc interchangeably. Line 34 states that oxidised zinc ores are being processed to a greater extent, yet in Line 35 emphasis is placed on the recovery of zinc oxide ores. These are two different mineral categories and Smithsonite is a zinc carbonate instead of a zinc oxide.
There are a couple of places where the word sulfurized appears, I assume the authors are referring to sulphidisation where NaSH, for example, is used?
In Lines 37-38 it is stated that non-sulphide zinc minerals are more difficult to float due to their greater reactivity. What do the authors mean by this? Sulphides can be highly reactive, especially Fe sulphides and when in contact with water form surface hydroxides which make them hydrophilic and difficult to float. Are the authors referring to the same mechanisms here?
Line 44, what is meant by lack of sulphide reactivity? Is this a reference to NaSH?
Line51, vulcanising, I assume the authors again mean sulphidising?
Line 54, as mentioned before zinc carbonate does not equal zinc oxide.
Reading the introduction it is not clear why HAA was selected as the collector for this study. What background information was sourced to determine that HAA may be a suitable collector for Smithsonite? Please provide more details as part of the introduction and literature.
2.0 Experimental
Line 65, "were derived" may also read "were obtained".
Line 70, "...(XRD) pattern analysis was shown..." should read "...(XRD) pattern is shown..."
Line 85, change to "...float concentrate and tailings".
The last sentence of Line 85 can be changed to: "All of the micro-flotation experiments were completed in triplicate.
Please provide some operating data for the micro-cell i.e. aeration rate, how were bubbles generated? Was frother used?
There is no mention as to why TOC studies were done only at pH 9. It only becomes clear when the float results are discussed later on in the paper. It may be a good idea to mention this here as well.
The last sentence in Line 100 can be changed to: "Each sample was measured in triplicate during zeta potential measurements.
With triplicate measurements, the authors may want to consider including error bars in Figure 5 as well.
Line 109, abundant mixing. Do the authors mean proper mixing?
As with the TOC section, an introduction as to why only pH 9 as selected will be useful to the reader at this stage.
Line 109 mentions that the sample was ground before FTIR analysis. In this process more clean surface area is exposed, will not affect the outcome of the FTIR measurements in that you measure fresh surface as well as collector coated surfaces?
For XPS analysis again please provide a brief explanation as to why only pH 9 was considered.
3.0 Results and discussion
Line 137 "variety" must read "varied".
Figure 4: the value of 1.67 indicated does not seem to reflect the differences between the before and after cases at 1.5x10-4 collector concentration. The difference appears to be closer to 3.5 or 4.0.
Line 165, equation 1 requires a reference.
Figure 7, split into two separate figures or ensure all images are on the same page.
References
There are two references with the number 39.
Overall
The authors are encouraged to see assistance with writing the paper in English.
Potential title review:
A mechanism for the adsorption of 2-(Hexadecanoylamino) Acetic Acid onto the surface of Smithsonite: Surface spectroscopy and micro-flotation experiments
Author Response
Reply to the review report is shown at PDF file

Reviewer 3 Report
Manuscript was printed and marked by reviewer. See Attachment.(Page 9 and after are not included).
Other Comments:
Provide the structure of HAA
Need to determine the pKa of HAA and ascertain and discuss its importance
Clarify what you mean by oxidized surface
In the experimental section, the titrate should have been washed with DI-water at pH 9
Need to reference recent flotation research such as LaDouceur et al (2018) and Noble (2017)

Author Response

(The authors gave the same response as above.)

Round 2
Reviewer 1 Report
The manuscript should be accepted for publication. Review revised sentences in Section 2.2 for grammatical errors and sentence fragments.